# Addressing the Role of Angiogenesis in Patients with Advanced Pancreatic Neuroendocrine Tumors Treated with Everolimus: A Biological Prospective Analysis of Soluble Biomarkers and Clinical Outcomes

**DOI:** 10.3390/cancers14184471

**Published:** 2022-09-15

**Authors:** Chiara Alessandra Cella, Francesca Spada, Alfredo Berruti, Francesco Bertolini, Patrizia Mancuso, Massimo Barberis, Eleonora Pisa, Manila Rubino, Lorenzo Gervaso, Alice Laffi, Stefania Pellicori, Davide Radice, Laura Zorzino, Angelica Calleri, Luigi Funicelli, Giuseppe Petralia, Nicola Fazio

**Affiliations:** 1Division of Gastrointestinal Medical Oncology and Neuroendocrine Tumors, European Institute of Oncology, IRCCS, 20141 Milan, Italy; 2Medical Oncology Unit, Department of Medical and Surgical Specialties, Radiological Sciences and Public Health, University of Brescia at the Azienda Socio Sanitaria Territoriale (ASST)-Spedali Civili, 25121 Brescia, Italy; 3Laboratory of Hematology-Oncology, European Institute of Oncology, IRCCS, 20141 Milan, Italy; 4Onco-Tech Lab, European Institute of Oncology IRCCS and Politecnico di Milano, 20019 Milan, Italy; 5Division of Pathology and Laboratory Medicine, European Institute of Oncology, IRCCS, 20141 Milan, Italy; 6Molecular Medicine Department, University of Pavia, 27100 Pavia, Italy; 7Medical Oncology and Hematology Unit, Humanitas Cancer Center, IRCCS Humanitas Research Hospital, 20089 Rozzano, Italy; 8Oncology Department, Azienda Ospedaliera di Lodi, 26900 Lodi, Italy; 9Division of Epidemiology and Biostatistics European Institute of Oncology, IRCCS, 20141 Milan, Italy; 10Division of Laboratory Medicine, European Institute of Oncology, IRCCS, 20141 Milan, Italy; 11Division of Diagnostic Haematopathology, European Institute of Oncology, IRCCS, 20141 Milan, Italy; 12Division of Medical Imaging and Radiation Sciences, European Institute of Oncology, IRCCS, 20141 Milan, Italy; 13SIRM, Italian College of Computed Tomography, Italian Society of Medical and Interventional Radiology, 20122 Milan, Italy; 14Department of Oncology and Hemato-Oncology, University of Milan, 20122 Milan, Italy; 15Precision Imaging and Research Unit, Department of Medical Imaging and Radiation Sciences, European Institute of Oncology, IRCCS, Via Ripamonti 435, 20141 Milan, Italy

**Keywords:** pancreatic neuroendocrine tumor, everolimus, angiogenesis, circulating cells, biomarkers

## Abstract

**Simple Summary:**

Despite the approval of new targeted therapies for pancreatic neuroendocrine tumors (PanNETs) over the past decades, the early identification of resistant tumors remains the major challenge, mainly because clear signs of tumor shrinkage are rarely achieved by imaging assessment. Starting from the hypothesis that angiogenesis can be implicated in the resistance to mTOR inhibitors, we evaluated a specific angiogenesis panel (through the measurement of soluble biomarkers for angiogenesis turnover, circulating endothelial cells, and circulating progenitors) as possible predictors of resistance to everolimus or everolimus efficacy in PanNETs. Our study showed that none of the investigated categories of biomarkers had a predictive value for everolimus resistance or efficacy. However, we suggest that circulating endothelial progenitors might be surrogate biomarkers for angiogenesis activity in PanNETs during everolimus treatment, and their baseline levels might correlate with survival outcomes. These data have never been reported before for NETs.

**Abstract:**

Background: The success of targeted therapies in the treatment of pancreatic neuroendocrine tumors has emphasized the strategy of targeting angiogenesis and the PI3K/AKT/mTOR pathway. However, the major challenge in the targeted era remains the early identification of resistant tumors especially when the efficacy is rarely associated to a clear tumor shrinkage at by imaging assessment. Methods: In this prospective study (NCT02305810) we investigated the predictive and prognostic role of soluble biomarkers of angiogenesis turnover (VEGF, bFGF, VEGFR2, TSP-1) circulating endothelial cells and progenitors, in 43 patients with metastatic panNET receiving everolimus. Results: Among all tested biomarkers, we found a specific subpopulation of circulating cells, CD31+CD140b-, with a significantly increased tumor progression hazard for values less or equal to the first quartile. Conclusion: Our study suggested the evidence that circulating cells might be surrogate biomarkers of angiogenesis activity in patients treated with everolimus and their baseline levels can be correlated with survival. However, further studies are now needed to validate the role of these cells as surrogate markers for the selection of patients to be candidates for antiangiogenic treatments.

## 1. Introduction

Pancreatic neuroendocrine tumors (PanNETs) are rare pancreatic neoplasms and represent less than 3% of primary pancreatic tumors [1]. Over the past decades, several therapies (other than somatostatin analogs), such as everolimus (EVE), sunitinib (SUN), and more recently, peptide receptor radionuclide therapy (PRRT), have been approved by the FDA and EMA for PanNETs based on pivotal trials [2,3,4]. Everolimus is an orally active mTOR inhibitor that has been reported to have anti-angiogenic properties distinct from vascular endothelial growth factor (VEGF) inhibitors [5]. In preclinical models, EVE has been shown to reduce the amount of mature and immature vessels, the total plasma, and VEGF in tumors without affecting blood vessel leakiness or tumor vascular permeability [5]. Some years later, this information was matched with clinical outcomes in a large biomarker analysis from the RADIANT-3 clinical trial. More in detail, Yao et al. proved that elevated baseline chromogranin A, neuron-specific enolase (NSE), placental growth factor (PlGF), and soluble VEGF receptor 1 (sVEGFR1) levels were found to be associated with a poor prognosis in patients with NETs receiving EVE [6]. Although some prognostic significance has been hypothesized, none of the components of the mTOR pathway have shown a reliable predictive value [7,8,9]. Moreover, the relative indolent behavior of NETs and the lack of sufficient discriminative power to monitor the effects of antivascular drugs make efficacy assessments even more challenging with the standard imaging techniques.

Therefore, in keeping with the concept that the early identification of responder patients is still an unmet clinical need in the targeted era, the role of angiogenesis as an adaptive prosurvival mechanism of tumor cells resistant to EVE deserves to be deeply investigated. Particularly, in the current study, we address the predictive and prognostic role of circulating biomarkers for angiogenesis turnover (BAT), as well as circulating cells (CCs), and we conduct a survival outcomes analysis. Hereby, we explain the rationale for the investigated biomarkers.

### 1.1. Biomarkers for Angiogenesis Turnover (BAT)

Angiogenesis is mediated by the balance between positive and negative regulators. Modulations in the expression of the following BAT have been proposed as direct/indirect biomarkers of anti-angiogenic drug activity:
VEGF is a strong growth factor that increases endothelial permeability. It can be released by cancer, stromal, and inflammatory cells, and it is stored in the platelets;Basic fibroblast growth factor (bFGF) is a pro-angiogenic growth factor released by tumor, stromal, and inflammatory cells and/or by mobilization from the extracellular matrix (ECM). It acts on endothelial cells via a paracrine mode of action; however, it can also be produced endogenously by endothelial cells via autocrine, intracrine, or paracrine modes, trigging angiogenesis signaling;VEGF receptor 2 (VEGFR2) is a member of the VEGFR family, and it is mainly localized in the vascular endothelium. VEGF ligands bind to VEGFR2, hence, triggering endothelial cell proliferation, survival, migration, and vascular permeability. Lastly, it contributes to angiogenesis activation;Thrombospondin (TSP1) is a family of five proteins involved in tissue remodeling associated with tumor cell proliferation and other physiological processes. It has been shown to suppress tumor growth by both inhibiting angiogenesis and activating transforming growth factor beta (TGF-β). Additionally, TSP1 exerts an anti-angiogenic effect through a direct effect on the migration of endothelial cells and the availability of VEGF.


### 1.2. Circulating Cells (CCs)

Circulating endothelial cells (CECs) are mature, differentiated endothelial cells shed from vessels during physiological endothelial turnover. They can be found in very small numbers within the blood of healthy individuals, and their number is indicative of and correlates with the degree of endothelial injury or dysfunction [10]. Circulating endothelial progenitors (CEPs) and pericyte progenitor cells (PPCs) are subsets of non-hematopoietic bone marrow-derived cells (BMDC) that are mobilized to complement local angiogenesis by acting as an alternative source of endothelial and mesenchymal cells [11]. In contrast to other bone BMDC types, CEPs and PPCs are thought to merge with the wall of a growing blood vessel, where they differentiate into mature endothelial and mesenchymal cells, thus, contributing to vessel growth [10,11]. Circulating mature endothelial cells (CECs) comprise: DNA (Syto16)+CD45-CD31+CD140bCD146+, including CD109+ and CD109-, and viable and apoptotic subpopulations. Circulating endothelial progenitors (CEPs) comprise: DNA (Syto16)+CD45-CD31+CD34+CD140b, including CD133+ and CD133-, and VEGFR2+ and VEGFR2- subpopulations. Circulating pericyte progenitors (PPCs) comprise: DNA (Syto16)+CD45-CD140b-, including CD31- subpopulations. To assess the blood-based biomarkers for angiogenesis that may predict the outcome of targeted therapies in cancer patients, many approaches have been tested in both preclinical and clinical studies; among these, the quantification of CECs and CEPs by flow cytometry has found wide application [12,13]. Increased plasma levels of CECs and CEPs have been reported in cancer patients. Modifications to their number and viability have shown predictive, prognostic, and dynamic biomarker value during patient selection and follow-up. Patients who responded to treatment with anti-angiogenic drugs showed clear changes in CEC and CEP levels when compared to baseline levels, while a subsequent increase predicted worse PFS [14,15,16]. At the time of this paper, no data regarding the predictive or prognostic role of these cells in patients with NETs were available, regardless of the therapeutic strategy.

In conclusion, EVE has been reported to have antivascular properties distinct from VEGF inhibitors. However, the role of blood-based biomarkers and circulating cells as direct or indirect indicators of angiogenesis activation and early predictors of EVE efficacy still needs to be clearly established in PanNETS.

## 2. Materials and Methods

This was a prospective clinical-biological study (clinicaltrials.gov: NCT02305810) including patients with well or moderately differentiated metastatic PanNETs (WHO, 2010 histology classification) who were treated with EVE and enrolled at the European Institute of Oncology between 2011 and 2016. This research has been approved by the local ethics committee (IEO S543/310). Patients with poorly differentiated neuroendocrine carcinoma, adenocarcinoid, goblet cell carcinoid, small cell carcinoma, and Merkel cell carcinoma were excluded from this study, as well as patients who received prior therapy with mTOR inhibitors.

### 2.1. Study Procedures

The written informed consent was signed and dated by the patients and investigators during the screening consultation. A clinical examination was scheduled at least monthly. Blood tests for CECs, CEPs, VEGF, bFGF, VEGFR2, and TSP1 were collected at baseline after one and three months of treatment, then at disease progression. A tissue biopsy at baseline, or optionally, at disease progression was required.

### 2.2. Sample Size

This is an exploratory study on the potential predictive and prognostic value of blood-based biomarkers (as direct or indirect indicators of angiogenesis activation) in patients with metastatic PanNETs treated with EVE. The two-tailed log-rank test ((α = 0.05, 1-β = 0.20) null hypothesis of HR = 0.30 (HR = Hazard Ratio), at three months from the start of treatment, for blood-based biomarkers values above the baseline median required 43 patients. The sample size was calculated to for the compensate the power loss of the log-rank test, assuming an average non-informative drop-out rate of 10%.

### 2.3. Statistical Analysis

The patients’ categorical variables were summarized by the count and percentage by mean and standard deviation (SD). BATs and CCs were summarized by the mean and interquartile range (IQR), and changes from the baseline were analyzed using repeated measures ANOVA. The time and the subjects’ ID entered the analysis as fixed and random factors, respectively. Patients whose time of visit did not fall within the range of ±10 days around the expected time, at 1 and 3 months after treatment started, were excluded from the analysis. Means comparison tests, with respect to the baseline, were adjusted for multiplicity using a simulation approach. Progression-free survival (PFS) and overall survival (OS) were defined as the time from EVE start to progression or death and the time from EVE start to death, respectively. OS and PFS risks, by the median cut-off value of the angiogenetic factors and CTC, were estimated using the Cox model; the resulting HRs were tabulated alongside 95% confidence intervals (CI). The median OS and PFS were estimated using the Kaplan–Meier method.

## 3. Results

Forty-three patients with histological diagnosis of well/moderately differentiated metastatic PanNETs were eligible and signed the informed consent for the study. The data analysis included 38 patients. Two were excluded due to screening failure (one due to a fast clinical deterioration and one due to thrombocytopenia); one patient was excluded due to an extreme irregularity of EVE assumption and, therefore, an unreliable correlation with the biological parameters; two patients were excluded due to an internal pathology review which did not confirm a well-differentiated tumor morphology. The mean age at diagnosis was 50 years (26–66). The median duration of treatment was 10.1 months. One-third of the patients had synchronous metastases. The complete baseline of the patient/tumor’s characteristics is summarized in Table 1.

Serum concentrations of BAT (VEGF, bFGF, VEGFR2, and TSP1) at the time from EVE start and their mean comparisons with the baselines at 1 and 3 months are shown in Figure 1. A number of significant changes were observed, except for in VEGF, for which the mean was significantly higher at 1 month (612 pg/mL vs. 448 pg/mL, *p* = 0.02) compared to the baseline, and VEGFR2, which showed a significant decreasing trend from the baseline (Appendix A).

The serum levels of CCs (CECs and CEPs) measured by the time from EVE start are shown in Table 2 (graphically represented in Appendix A). Among CECs, CD146+, vital CD146+, apoptotic CECs, and CD109+ subpopulations all significantly decreased for up to 3 months after the treatment started. Additionally, CD146+ (*p* = 0.01) and apoptotic CECs (*p* = 0.02) levels remained significantly lower than the baseline even at the progression timeline. Among pericyte precursors (progenitor perivascular cells, PPCs) and CEPs, the CD31-CD140b+ and Syto16+CD45dimCD34+ subpopulations mean counts were significantly lower at 1 and 3 months compared to the baseline, without any apparent trend. A significant lower mean compared to the baseline was observed for Syto16+CD45dimCD133+CD34+ at 3 months (*p* = 0.04) and for Syto16+CD45dimVEGFR2+ at 1 month (*p* = 0.007). Appendix A shows the CCs’ evaluation by flow cytometry.

### Survival Analysis

Median PFS and OS were 14.9 months (95% CI: (10.3–27.7) and 33.6 months (95% CI: (28.5—upper limit not estimable)), respectively. Kaplan–Meier curves for PFS and OS are shown in Figure 2. According to the BAT median cut-off at the baseline, no statistically significant hazard ratio (HR) was found for either PFS or OS, except for TPS1, which had a borderline significant (*p* = 0.04) OS risk reduction (HR = 0.33, 95% CI: 0.12–0.95) for TPS1 > 144 ng/mL (Table 3). Progression-free survival risk estimates according to the first (Q1), second (median), and third (Q3) quartiles of the baseline CCs are summarized as Hazard Ratios (HR) in Appendix A. After adjusting for multiple comparisons, only the PPC CD31-CD140b+ showed a significantly increased in PFS hazard for values less than or equal to the first quartile [Q1 = 51.4 counts/mL, HR = 3.78, 95% CI: (1.53–9.33), adjusted *p* = 0.01]. However, both the number of events and the subjects at risk were as few as eight and nine, respectively. No significant HRs were found for any other CCs, with the least significant hazard being for Syto16+CD45dim, CD133+CD34+ [Q1 = 87.4 counts/mL, HR = 2.70, 95% CI: (1.07–6.79), adjusted *p* = 0.06]. The Kaplan–Meier PFS curve, according to the baseline of the first quartile, for PPC CD31-CD140b+ is shown in Figure 3.

## 4. Discussion

In our study, neither soluble BAT, CECs, nor CEPs showed any predictive value for EVE efficacy. However, a specific subpopulation of circulating progenitors, CD31-CD140b+ (pericytes), was associated with a significantly shorter PFS when its values were less than or equal to the values in the first quartile.

Regarding BAT assessment, our study did not prove any prognostic nor predictive role, with the only exception being VEGFR2, which showed a significant decreasing trend over time. Similarly, TPS1, presented a borderline significant (*p* = 0.04) OS risk reduction. Conflicting results were previously reported regarding the implication of angiogenesis biomarker measurement (mainly focused on this discussion on VEGF and VEGFR2-3 values) in clinical practice, along with a more uncertain interpretation of their modulation over time [17,18,19,20]. Some robust data about the prognostic and clinico-pathological role of the tissue markers of angiogenesis were collected by Pinato et al. [21]. In their work, the clinical and follow-up information of 88 patients who underwent surgical treatment for gastro-entero-pancreatic neuroendocrine tumors (GEPNETs) were matched with histopathological features, such as vascular invasion and necrosis. Despite the identification that VEGFA expression correlated with the presence of liver metastases in the PanNet cohort, there was not any association demonstrated between VEGFA and OS. Furthermore, the majority of tumors displayed evidence of VEGFA expression, in line with the concept that GEPNETs are highly vascular tumors, and VEGFA expression and microvascular count seem to paradoxically reduce with progressive tumor de-differentiation in PanNETs [22]. Nonetheless, soluble biomarkers are likely more reliable predictive or prognostic drivers of angiogenesis turnover.

Surprisingly, in our analysis, the trend of VEGF and VEGFR2 levels seemed to show an inverse correlation, meaning that VEGF levels tended to increase while VEGFR2 concentrations decreased over time after EVE treatment started (from the baseline to 1- or 3-month timepoints). Similar findings have previously emerged with the antivascular agent SUN in different cancer settings. The first-in-human trial with SUN, including an analysis of the plasma levels of VEGF and sVEGFR2 at the baseline and after 28 days of treatment, showed a progressive increase in VEGF and a decrease in sVEGFR2 concentrations, demonstrating the on-target effects of the drug [23]. DePrimo et al. observed similar trends in metastatic renal cell carcinoma (mRCC) by the end of the first 4 weeks of SUN treatment, whereas the concentrations of soluble biomarkers tended to return close to baseline levels at the end of the first 2-week-off period. [24]. Consistent with the results observed in mRCC, Zurita found that high pre-treatment levels of sVEGFR2 were associated with longer OS in 107 patients with PanNETs and carcinoids, with higher sVEGFR2 concentrations in PanNETs compared to carcinoids. Patients with PanNETs also showed a trend toward higher baseline VEGF levels. Notably, at the end of the first cycle of sunitinib treatment (considering a 4-week schedule), an increase in VEGF levels and a decrease in sVEGFR2 and sVEGFR3 concentrations were observed [25].

Overall, these data demonstrate how far we are from interpreting any prognostic nor predictive role of baseline BAT in NETs, which may constitutively overexpress an angiogenic signature. Conversely, the modulation of soluble biomarkers of angiogenesis over time might be a surrogate endpoint of response to antivascular compounds. In our analysis, the inverse trends of soluble VEGF and VEGFR2 during EVE treatment might be consistent with a drug-related inhibitory effect on angiogenesis in patients with metastatic PanNETs.

Regarding the circulating cells (CCs), we found a specific subpopulation of circulating progenitors, CD31-CD140b+ (pericytes), with a significantly shorter PFS for values less than or equal to the first quartile. Conversely, none of the other CCs showed a significant predictive value for EVE activity. Despite the absence of any significant predictability, we found that the number of CCs (CECs and CEPs) decreased during EVE treatment, hence, corroborating the role of EVE in targeting angiogenesis. Former evidence suggested that pericytes could play an important part in tumor angiogenesis due to their ability to trigger the formation of abnormal microvessel networks embedding the tumor cells [26]. A long-standing in-home experience on the potential predictive or prognostic role of CCs has been reported by Bertolini et al., demonstrating that CECs, CEPs, and PPCs significantly increased in untreated cancer patients compared to healthy controls [10,13]. Additionally, they reached similar conclusions in a cohort of advanced breast cancer patients treated with metronomic chemotherapy, alone or in association with antivascular drugs, where the baseline CEC count was an indicator of efficacy [14,15,16]. Other results found in clear-cell RCC, which were reported by Cao et al., showed that an increased pericyte-generated microvessel formation conferred an anti-angiogenic resistance to treatments [26]. Paradoxically, a lowered pericyte population not only damaged the tumor vascular network, hence impairing tumor growth, but also increased the likelihood of metastatic dissemination [27,28]. In this sense, our analysis is in line with the above-mentioned data, suggesting that variations in the number and viability of PPCs (or committed pericytes) could provide relevant prognostic, but less likely predictive, information. No prior data about the correlation between CEP levels and EVE efficacy are available, except for a single preclinical study, where median values of CEPs were reduced by EVE monotherapy in severe human gastric cancer and a combined immunodeficient (SCID) mouse xenograft model. Despite the high variability in measurements, the decreasing trend of CEP levels under EVE monotherapy might always reflect the inhibitory effect on angiogenesis [29].

These data are also consistent with a previous experience gathered in gastro-entero-pancreatic (GEPNET) during SUN treatment, whereas expected, the number of CECs significantly decreased during the first 4 weeks of treatment as a consequence of an angiogenesis blockade. Conversely, no changes in CEPs were observed in the same study [25].

Our findings show that modulation of CEC, CEP, and PPC levels over time might represent an indirect measurement of the endothelial and pericyte turnover during EVE exposure, even though no predictive role can be established based on our analysis.

Our study presents several limitations. Firstly, although at the time of conceptualization it appeared timely, our study has been negatively affected by the long duration and high heterogeneity of the assessment, which invalidated a number of tests. Secondly, the timing of the sample collection was not strictly observed due to administrative delays and low compliance, as often happens in real-world evidence (RWE) studies. Therefore, this flexible management might have conditioned the reliability of statistical analyses.

On the other hand, a possible strength of our study is that a subpopulation of CCs (CD31-CD140b+) correlated with a significantly increased tumor progression hazard for values that were less than or equal to the first quartile, thus, demonstrating a prognostic value for CCs in PanNETs for the first time. Furthermore, the study population was quite homogeneous for types of treatment (EVE) and primary sites (PanNETs).

Finally, although other drugs with preponderant antivascular effects could have been more suitable for our study, the initial hypothesis that angiogenesis might be ascribed as a mechanism for resistance to EVE treatment remains original and innovative, and it deserves to be rigorously investigated, as already addressed in a previous literature review [30]. EVE has been shown to exert anti-angiogenic activity by both direct effects on vascular cell proliferation and indirect effects on growth factor production, with in vitro evidence in colon, breast, renal, melanoma, cervical, and glioma cell lines. However, reports on the activity of EVE during the early stages of in vitro vasculogenesis and angiogenesis in NETs need to be further addressed [31].

## 5. Conclusions

In conclusion, our study did not provide conclusive results about the predictive role of EVE resistance or the efficacy of biomarkers for angiogenic turnover/activity. However, we reported that the baseline count of CCs (CEP) might represent an indirect measure of endothelial and pericyte turnover and, consequently, can be advocated as a surrogate biomarker of angiogenesis activation. Intriguingly, the hypothesis generated by our study needs to be further investigated in other homogenous populations (e.g., extrapancreatic NENs) treated with EVE.

## Figures and Tables

**Figure 1 cancers-14-04471-f001:**
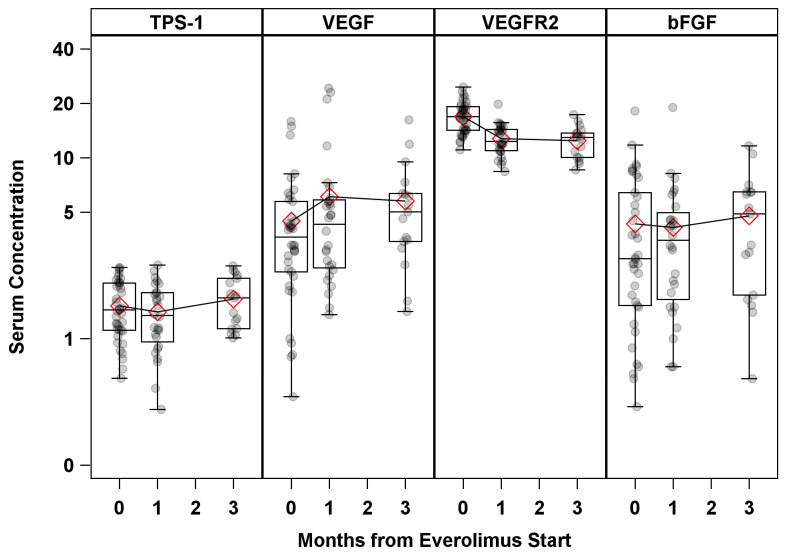
Serum BAT concentration by time from Everolimus start. Serum concentration units: TPS1 (ng/mL) × 100; VEGF (pg/mL) × 100; VEGFR2 (pg/mL) × 100; bFGF (pg/mL).

**Figure 2 cancers-14-04471-f002:**
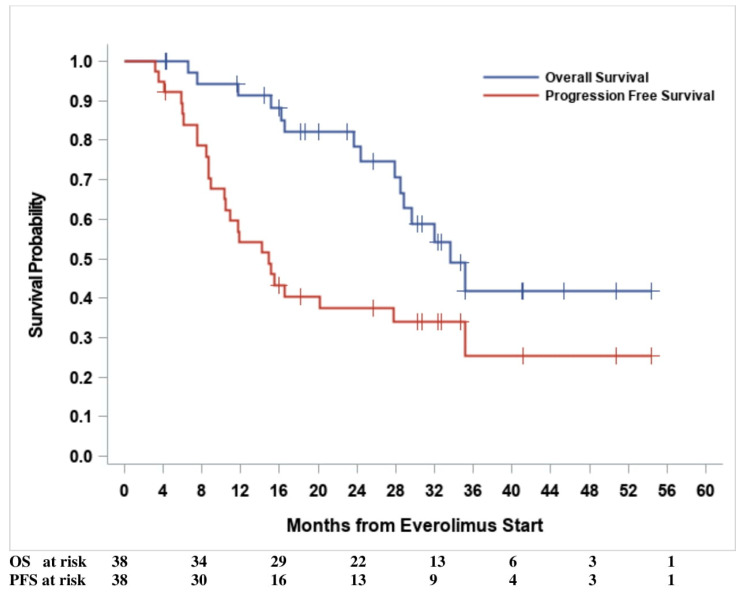
Survival outcomes.

**Figure 3 cancers-14-04471-f003:**
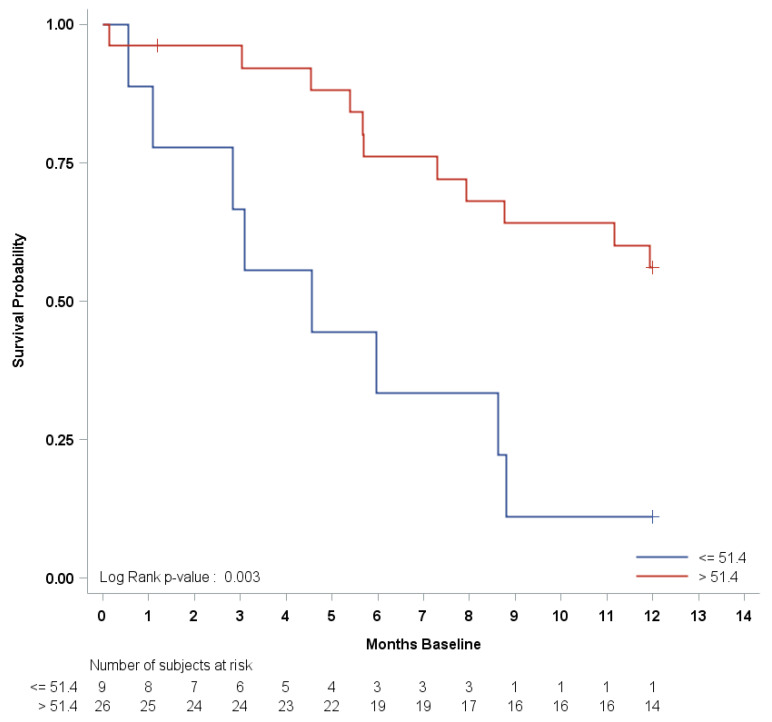
Progression-free survival Kaplan–Meier curve by first quartile of baseline PPC CD31-CD140b+.

**Table 1 cancers-14-04471-t001:** Patient’s characteristics and clinical course, *N* = 38.

Characteristic	Statistics ^1^
Age (years) at	
Diagnosis	50.0 (10.1) ^2^
Everolimus start	54.4 (10.2)
Metastases	
Synchronous	29 (76.3)
Metachronous	9 (23.7)
Ki 67 (%)	
(<3)	1 (2.7)
(3–20)	31 (81.5)
(21–55)	5 (13.1)
*missing*	*1 (2.7)*
Sex	
Male	19 (50.0)
Female	19 (50.0)
**Baseline** ^ **68** ^ **Ga-PET/CT**	32 (84.2)
Previous Treatments ^3^	
Liver-directed treatments	10 (26.3)
Chemotherapy	11 (29.0)
Peptide Receptor Radionuclide Therapy (PRRT)	20 (52.6)
Somatostatin Analogs (SSA)	30 (78.9)
Sunitinib	5 (13.1)
Surgery	
primary site	14 (36.8)
metastatic site	2 (5.3)
primary and metastatic site	9 (23.7)
none	13 (34.2)
Functionally active tumors	
Yes	5 (13.2)
No	33 (86.8)

^1^ Statistics are: Mean (SD) for Age, ^2^ (min = 26, max = 66), *N* (%) otherwise, ^3^ Not mutually exclusive treatments; SD = Standard Deviation.

**Table 2 cancers-14-04471-t002:** Summary Statistics of CCs by time from everolimus start.

	Time	*N*	Mean (IQR) ^1^	Adj *p*-Value ^2^
CEC CD146+	Baseline	36	119 (67.7–163)	
	Month 1	37	64.6 (31.9–73.7)	<0.001
	Month 3	34	54.8 (23.4–66.0)	<0.001
	at PD	13	52.4 (22.8–66.5)	0.01
CEC, Apo (%)	Baseline	36	51.3 (39.0–68.0)	-
	Month 1	37	59.5 (38.0–80.0)	0.34
	Month 3	34	57.6 (42.0–70.0)	0.59
	at PD	13	52.9 (46.0–67.0)	1.00
CEC CD146+ Vital	Baseline	36	61.0 (24.5–84.3)	-
	Month 1	37	31.4 (6.3–36.7)	0.01
	Month 3	34	26.9 (7.3–30.6)	0.003
	at PD	13	23.3 (11.4–33.4)	0.05
Apoptotic CEC	Baseline	35	59.6 (28.3–68.0)	-
	Month 1	36	33.7 (16.4–42.8)	0.002
	Month 3	33	28.6 (13.6–38.9)	<0.001
	at PD	13	29.2 (11.4–33.3))	0.02
CD140b+ pericytes	Baseline	36	22.4 (7.6–30.8)	-
	Month 1	37	15.6 (0.0–16.2)	0.50
	Month 3	33	13.2 (2.5–14.7)	0.25
	at PD	13	11.4 (3.6–16.5)	0.36
CEC CD109+	Baseline	34	111 (50.4–160)	-
	Month 1	37	51.5 (21.6–71.3)	0.003
	Month 3	34	51.3 (21.0–54.1)	0.004
	at PD	13	50.9 (19.6–75.0)	0.05
PPC CD31-CD140b+	Baseline	36	107 (51.4–153)	-
	Month 1	37	49.9 (27.361.8)	0.001
	Month 3	33	62.0 (26.8–73.5)	0.02
	at PD	13	53.1 (24.9–75.6)	0.04
Syto16+CD45dimCD34+	Baseline	36	729 (401–909)	-
	Month 1	37	481 (226–601)	0.002
	Month 3	33	534 (256–711)	0.01
	at PD	12	537 (186–611)	0.33
Syto16+CD45-CD34+	Baseline	36	52.3 (27.8–67.2)	-
	Month 1	37	36.7 (23.5–44.8)	0.56
	Month 3	33	55.2 (25.057.6)	0.99
	at PD	12	56.8 (37.5–63.7)	0.99
Syto16+CD45dimCD133+CD34+	Baseline	36	213 (87.4–269)	-
	Month 1	37	164 (62.3–234)	0.46
	Month 3	33	126 (67.6–160)	0.04
	at PD	12	157 (46.2–295)	0.66
Syto16+CD45dimVEGFR2+	Baseline	36	6.57 (0.00–9.25)	-
	Month 1	37	1.90 (0.00–2.50)	0.007
	Month 3	32	2.84 (0.00–4.78)	0.06
	at PD	11	2.11 (0.00–3.55)	0.14

^1^ IQR = Interquartile Range; ^2^ Repeated Measures Adjusted *p*-values for Multiple comparisons vs. Baseline.

**Table 3 cancers-14-04471-t003:** Progression-free survival and overall survival risk estimates according to BAT median cut-off values at baseline.

		Cut-Off (Median)	No. Failures /at Risk	Hazard Ratio (95% CI)	Adj *p*-Value
PFS	VEGF (pg/mL)	≤365	12/19	Ref	
		>365	13/19	1.06 (0.48–2.33)	0.88
	VEGF R (pg/mL)	≤1689	12/19	Ref	
		>1689	13/19	1.30 (0.59–2.85)	0.52
	BFGF (pg/mL)	≤2.8	15/19	Ref	
		>2.8	10/19	0.50 (0.22–1.12)	0.09
	TPS1 (ng/mL)	≤144	14/19	Ref	
		>144	11/19	0.65 (0.29–1.44)	0.29
OS	VEGF (pg/mL)	≤365	8/19	Ref	
		>365	8/19	1.05 (0.39–2.79)	0.93
	VEGF R (pg/mL)	≤1689	9/19	Ref	
		>1689	7/19	0.61 (0.23–1.66)	0.33
	BFGF (pg/mL)	≤2.8	8/19	Ref	
		>2.8	8/19	0.60 (0.22–1.62)	0.31
	TPS1 (ng/mL)	≤144	10/19	Ref	
		>144	6/19	0.33 (0.12–0.95)	0.04

## Data Availability

This study did not report any data.

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
