# Peer review of "Addressing the Role of Angiogenesis in Patients with Advanced Pancreatic Neuroendocrine Tumors Treated with Everolimus: A Biological Prospective Analysis of Soluble Biomarkers and Clinical Outcomes"

_cancers, 2022, doi:10.3390/cancers14184471_

Round 1
Reviewer 1 Report
The authors present an interesting study assessing biomarkers that could be used for evaluating patients responding to EVE. Their findings show an interesting inverse relationship between the expression of soluble VEGF an VEGFR2. They have identified a subpopulation of pericytes that correlated with shorter PFS.
I have a few minor comments:
a) The association between CD31+/CD140b- cells and short PFS is interesting. The authors could expand a bit in their discussion as to why they think this is. Does it impact angiogenesis? Has this been reported in other cancer models?
b)Line 168: Did the remaining patients develop metastases at a later or earlier stage?
Author Response
The authors present an interesting study assessing biomarkers that could be used for evaluating patients responding to EVE. Their findings show an interesting inverse relationship between the expression of soluble VEGF an VEGFR2. They have identified a subpopulation of pericytes that correlated with shorter PFS.
I have a few minor comments:
a) The association between CD31+/CD140b- cells and short PFS is interesting. The authors could expand a bit in their discussion as to why they think this is. Does it impact angiogenesis? Has this been reported in other cancer models? Author's reply: This issue has been addressed according to the available evidence for breast, renal and neuroendocrine tumors. As reported in ref 11, NET is one of the cancer types were PPCs were significantly increased and likely have a crucial role in supporting disease progression. We have also reported data about the correlation between CEP levels and EVE efficacy in a human gastric cancer severe combined immunodeficient (SCID) mouse xenograft model (line 272-296). In the conclusive discussion, it has been postulated that CEPs levels might change as a consequence of an anti-vascular drug-related effect, but how this affects survival need to be further explored.
b) Line 168: Did the remaining patients develop metastases at a later or earlier stage? Author's reply: All patients had liver metastases at the beginning of everolimus treatment.
Reviewer 2 Report
I read with interest the paper by Cella et al. on PanNET with metastasis who received anti-angiogenic therapy (other than VEGF inhibitors) such as Everolismus. The authors included in this prospective study 43 patients and measured different parameters at the start of Everolimus as well as 1 and 3 months after treatment initiation. The authors correlated these data (soluble biomarkers and circulating hematopoietic cells) with overall and progression free survival (PFS). The authors found no correlation with oncological outcomes regarding soluble biomarkers. Only the cell population CD31+CD140b+ was able to stratify PFS after adjustment.
I have some comments:
Is there a difference if you investigate these effects only in patients with active/inactive tumors?
Please correct all dashes in the manuscript which are somehow mixed up
Please correct the abbreviation “OS” for PFS in line 155
Please correct “of” in line 122
Please correct in line 48 “might correlate”
Finally, I suggest some editing of the manuscript.
Author Response
I read with interest the paper by Cella et al. on PanNET with metastasis who received anti-angiogenic therapy (other than VEGF inhibitors) such as Everolismus. The authors included in this prospective study 43 patients and measured different parameters at the start of Everolimus as well as 1 and 3 months after treatment initiation. The authors correlated these data (soluble biomarkers and circulating hematopoietic cells) with overall and progression free survival (PFS). The authors found no correlation with oncological outcomes regarding soluble biomarkers. Only the cell population CD31+CD140b+ was able to stratify PFS after adjustment.
I have some comments:
Is there a difference if you investigate these effects only in patients with active/inactive tumors? Author's reply: The clinical and histopathological characteristics of pancreatic tumors are quite homogenous in our study population and therefore can be considered as active cancers.
Please correct all dashes in the manuscript which are somehow mixed up Author's reply: Modified in the text
Please correct the abbreviation “OS” for PFS in line 155 Author's reply: Modified in the text
Please correct “of” in line 122 Author's reply: Modified in the text
Please correct in line 48 “might correlate” Author's reply: Modified in the text
Finally, I suggest some editing of the manuscript. Author's reply: The manuscript has been re-edited.
Reviewer 3 Report
The manuscript by Cella et al describes correlative studies that accompanied NCT02305810, a study of metastatic panNET patients treated with the mTOR inhibitor everolimus. The premise was that biomarkers of angiogenic turnover (BAT) and/or circulating cells may predict the efficacy of EVE as an anti-angiogenic/anti-tumor agent. Although no definitive or predictive results were obtained, it is important that studies such as this make it into the literature to help inform oncological drug development.
Since there are some interesting results with the levels of circulating cells, this manuscript would be greatly improved by a better description of the type or status of the cells that correspond to the flow cytometry markers shown in in Table 2. For example, section 1.2 (lines 100-124) and/or Table 2 could include a key that would help the reader interpret the difference between CD146+, CD146+vital, CD109+,etc.
It would be helpful if the Summary specifically called attention to circulating endothelial cell and endothelial cell progenitors as being a circulating cell type that may provide prognostic information in this study. The reference simply to ‘circulating cells’ may be interpreted as circulating tumor cells, which were not measured in this study whereas there is detailed information on circulating endothelial cells and progenitors.
The inverse trend of plasma VEGF and VEGFR2 during EVE treatment is interesting but the authors go too far in suggesting it might a “proof” of a drug-related inhibitory effect of angiogenesis (line 265) and should instead say it is suggestive or consistent with this effect.
Minor comments:
Hematopoietic and not- cells (lines 44 and 220) is not informative and better terminology should be used.
Progression Free Survival (PFS), line 155
There are numerous erroneous hyphenations throughout that need correction
Author Response
The manuscript by Cella et al describes correlative studies that accompanied NCT02305810, a study of metastatic panNET patients treated with the mTOR inhibitor everolimus. The premise was that biomarkers of angiogenic turnover (BAT) and/or circulating cells may predict the efficacy of EVE as an anti-angiogenic/anti-tumor agent. Although no definitive or predictive results were obtained, it is important that studies such as this make it into the literature to help inform oncological drug development.
Since there are some interesting results with the levels of circulating cells, this manuscript would be greatly improved by a better description of the type or status of the cells that correspond to the flow cytometry markers shown in in Table 2. For example, section 1.2 (lines 100-124) and/or Table 2 could include a key that would help the reader interpret the difference between CD146+, CD146+vital, CD109+, etc.
Author's reply: As now reported in line 116-121, we have clarified that circulating, mature and progenitor cells measured in the study included
Circulating mature endothelial cells (CECs) DNA(syto16)+CD45-CD31+CD140b-CD146+, incuding CD109+ and CD109-, viable and apoptotic subpopulations.
Circulating endothelial progenitors (CEPs) DNA(Syto16)+CD45-CD31+CD34+CD140b-, incuding CD133+ and CD133-, VEGFR2+ and VEGFR2- subpopulations.
Circulating pericyte progenitors (PPCs) DNA(Syto16)+CD45-CD140b-, including CD31- subpopulations
Manuscript was additionally modified (line 106-121) as following:
Circulating endothelial progenitors (CEPs) and pericyte progenitor cells (PPCs) are subsets of non-hematopoietic bone marrow-derived cells (BMDC) that are mobilized to complement local angiogenesis by acting as an alternative source of endothelial and mesenchymal cells [11]. In contrast to all other bone BMDC types that reside at the perivascular site, CEPs and PPCs are thought to merge with the wall of a growing blood vessel, where they differentiate into mature endothelial and mesenchymal cells, and thus can contribute to vessel growth [10,11]. To assess blood-based biomarkers of angiogenesis that may predict outcome to targeted therapies in cancer patients, many approaches have been tested in both pre-clinical and clinical studies and - among these - the quantification of CECs and CEPs by flow cytometry has found wide application [12,13]. Increased plasma levels of CECs and CEPs have been reported in cancer patients and their modifications in number and viability has shown predictive, prognostic and dynamic biomarker value for patients’ selection and follow-up. Patients who responded to treatment with anti-angiogenic drugs show clear changes of CECs and CEPs levels comparing to baseline and appear to predict response to treatment, while a subsequent increase predicts worse PFS [14-16]. By the time of this paper, no data are available regarding the predictive or prognostic role of these cells in patients with NETs, regardless the therapeutic strategy.
It would be helpful if the Summary specifically called attention to circulating endothelial cell and endothelial cell progenitors as being a circulating cell type that may provide prognostic information in this study. The reference simply to ‘circulating cells’ may be interpreted as circulating tumor cells, which were not measured in this study whereas there is detailed information on circulating endothelial cells and progenitors. Author's reply: Modified in the text
The inverse trend of plasma VEGF and VEGFR2 during EVE treatment is interesting but the authors go too far in suggesting it might a “proof” of a drug-related inhibitory effect of angiogenesis (line 265) and should instead say it is suggestive or consistent with this effect. Author's reply: Modified in the text
Minor comments:
Hematopoietic and not- cells (lines 44 and 220) is not informative and better terminology should be used. Author's reply: Modified in the text
Progression Free Survival (PFS), line 155 Author's reply: Modified in the text
There are numerous erroneous hyphenations throughout that need correction Author's reply: Modified in the text
Reviewer 4 Report
In the manuscript by Cella et al. has defined the function of angiogenesis in advanced pancreatic neuroendocrine tumor patients, who are treated with Everolimus. Further, the author has tried to identify a specific marker that can be used as a predictor of resistance or efficacy to everolimus in pancreatic neuroendocrine tumors. The author has shown some correlation between the circulating cells and angiogenesis in pancreatic neuroendocrine tumor under everolimus treatment. The manuscript is well written; however, I have few suggestions:
1. Please correct the words, like: neu-roendocrine, asso-ciated to neuroendocrine, associate. These kind of mistake with other words also is present in throughout the manuscript. Please fix the texts for these mistakes.
2. Abstract: What is the circulating cells? please mention that in abstract otherwise, it looks very vague.
3. Figure 2 and 3: Please mention the p value and the number of patients in the figures. It will give more clarity to the readers.
Author Response
In the manuscript by Cella et al. has defined the function of angiogenesis in advanced pancreatic neuroendocrine tumor patients, who are treated with Everolimus. Further, the author has tried to identify a specific marker that can be used as a predictor of resistance or efficacy to everolimus in pancreatic neuroendocrine tumors. The author has shown some correlation between the circulating cells and angiogenesis in pancreatic neuroendocrine tumor under everolimus treatment. The manuscript is well written; however, I have few suggestions:
- Please correct the words, like: neu-roendocrine, asso-ciated to neuroendocrine, associate. These kind of mistake with other words also is present in throughout the manuscript. Please fix the texts for these mistakes. Author's reply: Modified in the text
- Abstract: What is the circulating cells? please mention that in abstract otherwise, it looks very vague. Author's reply: In the conclusive sentence of the abstract “Circulating cells” has been replaced by “circulating progenitors” (line 48)
- Figure 2 and 3: Please mention the p value and the number of patients in the figures. It will give more clarity to the readers. Author's reply: P-value in figure 2 does not apply since the curves represents two different outcomes (PFS and OS) on the very same patients . Figure 3 has the p-values embedded within and the number of subjects in the footnote as requested.